# Long Short-Term Memory Neural Network with Transfer Learning and Ensemble Learning for Remaining Useful Life Prediction

**DOI:** 10.3390/s22155744

**Published:** 2022-08-01

**Authors:** Lixiong Wang, Hanjie Liu, Zhen Pan, Dian Fan, Ciming Zhou, Zhigang Wang

**Affiliations:** 1National Engineering Research Center of Fiber Optic Sensing Technology and Networks, Wuhan University of Technology, Wuhan 430070, China; wanglixong@foxmail.com (L.W.); liuxiaoniu@whut.edu.cn (H.L.); panloveing@foxmail.com (Z.P.); fandian@whut.edu.cn (D.F.); 2School of Mechanical and Electrical Engineering, Wuhan University of Technology, Wuhan 430070, China; 3School of Machinery and Automation, Wuhan University of Science and Technology, Wuhan 430081, China; wzhigang@wust.edu.cn

**Keywords:** remaining useful life, deep learning, health indicator, transfer learning, ensemble learning

## Abstract

Prediction of remaining useful life (RUL) is greatly significant for improving the safety and reliability of manufacturing equipment. However, in real industry, it is difficult for RUL prediction models trained on a small sample of faults to obtain satisfactory accuracy. To overcome this drawback, this paper presents a long short-term memory (LSTM) neural network with transfer learning and ensemble learning and combines it with an unsupervised health indicator (HI) construction method for remaining-useful-life prediction. This study consists of the following parts: (1) utilizing the characteristics of deep belief networks and self-organizing map networks to translate raw sensor data to a synthetic HI that can effectively reflect system health; and (2) introducing transfer learning and ensemble learning to provide the required degradation mechanism for the RUL prediction model based on LSTM to improve the performance of the model. The performance of the proposed method is verified by two bearing datasets collected from experimental data, and the results show that the proposed method obtains better performance than comparable methods.

## 1. Introduction

In mechanical equipment, rolling element bearings are one of the most critical and vulnerable components due to their transferring motion and withstanding loads during the operation of machinery [1,2]. To guarantee the normal operation of machinery and to avoid catastrophic events, condition-based maintenance (CBM) or predictive maintenance (PM) need to be arranged using remaining-useful-life (RUL) predictions of bearings [3]. Thus, the accuracy of RUL prediction is greatly significant for the safety and reliability of manufacturing equipment.

With increasing complexity of mechanical equipment, traditional RUL approaches are unable to suit the correspondingly higher standards and higher levels for mechanical equipment. Therefore, data-driven RUL methods have been widely studied in recent years [4,5]. The framework of data-driven methods mainly consists of data acquisition, construction of health indicators (HI) and RUL prediction [6]. According to a recent review, deep learning has attracted increasing attention in the field of RUL prediction. Lei et al. [7] utilized a multi-scale dense-gate recurrent neural network to identify relevant information at different timescales to improve the performance of model. Yang et al. [8] proposed an improved long-short term memory neural network to estimate bearing performance degradation. In [9], a hybrid model based on long short-term memory and Elman neural networks predicted RUL for lithium-ion batteries. Despite existing prognosis methods achieving satisfactory accuracy, there are still two deficiencies that limit their industrial extension and application.

The first shortcoming is that most HI construction methods adopt hand-designed labels for supervised training, which makes HI construction time-consuming and laborious. A deep belief network (DBN) is a deep learning method [10] that can extract the most-suitable feature representations from raw data due to its nonlinear network structure. It has been gradually applied in the field of prognostic and health management (PHM) [11]. Peng et al. [12] utilized deep belief networks with an unsupervised algorithm to construct HI and achieved RUL prediction by the improved particle filter. Xu et al. [13] proposed an improved unsupervised deep belief network to estimate bearing performance degradation.

The second shortcoming is that RUL prediction models are trained based on the availability of large training datasets with fault trends. However, in real conditions, we can only obtain relatively small sets of fault data for the predicted target. This makes it difficult to acquire high accuracy in RUL prediction. To overcome this drawback, transfer learning and ensemble learning are provided, solving the problem of insufficient accuracy due to lack of historical measurement data. Transfer learning uses the knowledge learned from the source domain to improve learning in the target domain. Ensemble learning is a learning approach in which results are combined by multiple learning algorithms to improve performance. In the field of CBM, transfer learning and ensemble learning have been applied mainly to fault diagnosis, but there are a few RUL prediction applications. For instance, Shen et al. [14] proposed deep convolutional neural networks with ensemble learning and transfer learning to estimate the capacity of lithium-ion batteries. Zhang et al. [15] presented an instance-based ensemble deep transfer learning network to recognize degradation of ball screws. In [16], transfer component analysis (TCA) was used to find the common feature representation between different bearings, and then an SVM prediction model was constructed to achieve RUL prediction. Zhang et al. [17] utilized an ensemble learning-based prognostic approach to improve the performance of models in RUL prediction of aircraft engines.

To overcome the aforementioned drawbacks, this paper presents a hybrid model of HI construction based on a deep belief network (DBN) and a self-organizing map (SOM), and an RUL prediction model based on a long short-term memory neural network (LSTM) with transfer learning and ensemble learning. As shown in Figure 1, HI is first constructed from measured vibration data by the hybrid model based on DBN and SOM. Following that, a long short-term memory neural network with transfer learning and ensemble learning is constructed to improve the performance RUL prediction of target bearing by using the auxiliary data of the inherent degradation trend under different working conditions. In the end, the performance of the proposed method is demonstrated by experimental bearing datasets. The main contributions of this paper are summarized as follows.
(1)Unsupervised HI construction based on DBN and SOM is used to construct HI. The process of feature extraction and feature fusion does not require artificial labels. Further, there is no need to determine a fault threshold (FT) based on the experience of researchers. Therefore, the constructed HI can eliminate the influence of personnel participation.(2)Transfer learning and ensemble learning are introduced into a long short-term memory neural network to improve the accuracy and robustness of the model when it is trained with a small amount of data. Experimental results indicate that LSTM with transfer learning and ensemble learning (LSTM-ETL) has better performance than the LSTM.

The rest of the paper is organized as follows: Section 2 reviews existing studies related to the basic theory. In Section 3, the proposed method is introduced. In Section 4, the performance of the proposed method is validated through an experimental dataset. Finally, conclusions are drawn in Section 5.

## 2. Related Basic Theory

### 2.1. Deep Belief Network

A deep belief network is stacked by multiple restricted Boltzmann machines (RBMs); as a deep learning method, it can extract essential patterns from raw data based on its strong nonlinear mapping performance [18]. For instance, Peng et al. [12] proposed a deep belief network to construct HI and employed an improved particle filter model for RUL prediction.

In a deep belief network, the training process is divided into an unsupervised pre-training phase and a supervised fine-tuning phase. In the pre-training phase, the output of the previous RBM is invoked as the input of the next RBM to update network parameters through a contrastive divergence (CD) algorithm. In the fine-tuning phase, a back-propagation (BP) algorithm is employed to update network parameters from the bottom layer to the top layer, as shown in Figure 2.

### 2.2. Long Short-Term Memory Neural Network

Compared with a traditional feed-forward neural network, a recurrent neural network (RNN) is a deep learning model into which a notion of time has been introduced. It is able to relate time and data. As the time sequence increases, the RNN [19] structure deepens by hidden-layer back-propagation. RNNs can have problems of gradient explosions or gradient vanishing in the training process.

Therefore, for long-time-sequence data, RNNs cannot remember the relationship between the current information and the long-time information in the time sequence. To solve these kinds of problems, a memory cell is used in LSTM architecture. As shown in Figure 3, the memory cell replaces the hidden layer of a traditional RNN, and it contains three kinds of gate structure: the forget gate, the input gate, and the output gate. Therefore, in time sequences, LSTMs are more suitable than traditional RNN for predicting data.

The input of LSTM consists of the previous time output $h_{t-1}$ and the current time input $x_{t}$, which is input to the memory cells to determine the information discarded by the forget gate. The forget gate $f_{t}$ controls what information from the previous memory cells is to be discarded, as defined by:(1)$$f_{t}=\sigma \left(w_{\mathrm{fx}}x_{t}+w_{\mathrm{fh}}h_{t-1}+b_{f}\right)$$
where σ indicates the sigmoid function; $w_{\mathrm{fx}}$ and $w_{\mathrm{fh}}$ indicate the weight from the input layer to the hidden layer of the forget gate and the weight from the previous hidden layer to the hidden layer of the forget gate at time *t*, respectively; and $b_{f}$ is the bias of the forget gate.

Then, the memory cells need to determine the updated new information combined with the input gate $i_{t}$, the candidate value $g_{t}$, the forget gate $f_{t}$, and the previous state $c_{t-1}$. The following formula can be used to describe the process in which the memory cells determine the updated information.
(2)$$i_{t}=\sigma \left(w_{\mathrm{ix}}x_{t}+w_{\mathrm{ih}}h_{t-1}+b_{i}\right)$$
(3)$$g_{t}=\sigma \left(w_{\mathrm{gx}}x_{t}+w_{\mathrm{gh}}h_{t-1}+b_{g}\right)$$
(4)$$c_{t}=f_{t}⊗c_{t-1}+i_{t}⊗g_{t}$$
where $w_{\mathrm{ix}}$ and $w_{\mathrm{ih}}$ indicate the weight from the input layer to the hidden layer of the input gate and the weight of the previous hidden layer to the hidden layer of the input gate at time *t*, respectively; $b_{i}$ and $b_{g}$ are the bias of the input gate and the candidate gate, respectively; $c_{t}$ indicates the cell state at the present time; tanh(·) indicates a hyperbolic tangent function; and ⊗ indicates element-wise multiplication.

Finally, in the memory cells, the network decides the output by the current state $c_{t}$ and the output gate $o_{t}$, which is defined as:(5)$$o_{t}=\sigma \left(w_{\mathrm{ox}}x_{t}+w_{\mathrm{oh}}h_{t-1}+b_{o}\right)$$
(6)$$h_{t}=o_{t}⊗\tanh(c_{t})$$
where $w_{\mathrm{ox}}$ and $w_{\mathrm{oh}}$ indicate the weight from the input layer to the hidden layer of the output gate and the weight of the previous hidden layer to the hidden layer of the output gate at time *t* respectively; and $b_{o}$ is the bias of the output gate.

## 3. Methodology

### 3.1. Proposed HI Construction

It is well-known that the performance of bearings gradually decreases with time. Correspondingly, to represent the health state of a bearing, an effective and universal method of HI construction is needed. This section mainly provides the procedure for HI construction based on unsupervised learning. The key idea is that the extracted features should retain the essential information of the raw data. Since DBN has strong nonlinear mapping performance, it is able to extract essential patterns from raw data. With data collected from *n* time steps T, the collected data T is firstly preprocessed to [0, 1] by normalizing according to the corresponding formula $t^{*}=\left(t-t_{\min}\right)/\left(t-t_{\max}\right)$, where t is raw data, $t_{\min}$ is the minimum value of raw data, and $t_{\max}$ denotes the maximum value of raw data. After the DBN structure is determined, the normalized training dataset $T^{*}$ is employed to train DBN. Then, the trained DBN is obtained by the pre-training phase and the fine-tuning phase using the normalized dataset. Finally, the extracted features $h_{n}$ are expressed as follows:(7)$$\{\begin{array}{c} h_{1}=\sigma \left(W_{1}t+b_{1}\right)\\ h_{2}=\sigma \left(W_{2}t_{1}+b_{2}\right)\\ \begin{array}{c} \vdots \\ . \end{array}\\ h_{n}=\sigma \left(W_{n}t_{n-1}+b_{n}\right) \end{array}$$
where $W_{i}$, $b_{i}$ are parameters of DBN (*i* = 1, 2,…, *n*), *x* denotes the raw vibration signal, σ denotes the activation function of DBN, n is the number of hidden layers, and $h_{i}$ denotes the output of the *i*th hidden layer.

To fully utilize the feature information extracted from the raw data, it is necessary to fuse the features through an appropriate method. A self-organizing map is an unsupervised algorithm that is only composed of an input layer and a competitive layer; it can map high-dimensional data into a two-dimensional topology structure. Therefore, to take advantage of degraded information from the extracted features, an SOM is employed to fuse the extracted features. After the SOM structure is determined, the extracted training feature $h_{n}$ is employed to train the SOM. Then, the trained SOM is obtained by a competitive formula of training features. Finally, the feature-set is feed into the trained SOM to construct HI according to the corresponding formula $h_{i}=\|f-m_{BMU}\|$, where $h_{i}$ is the HI at time *i*, f denotes the large features of input, and $m_{BMU}$ stands for the weight vector of the best matching unit (BMU) of input f [20]. The flowchart of the proposed HI construction based on unsupervised learning is shown in Figure 4.

### 3.2. Model Construction Process

#### 3.2.1. LSTM Parameter Transfer

As mentioned before, the LSTM algorithm has the inevitable problem of low prediction accuracy with small sets of fault data. Hence, we introduce learned knowledge from the source domain to improve the performance of LSTM in the target domain. The source dataset is usually different but related to the target domain. Based on the HI constructed in Section 3.1, the source dataset and the target dataset are expressed as follows:(8)$$Y_{s}=\left\{HI_{1}^{s},HI_{2}^{s},\ldots ,HI_{n}^{s}\right\}$$
$$Y_{t}=\left\{Y_{t}{}^{train},Y_{t}{}^{test}\right\}\;\mathrm{with}\;\{\begin{array}{c} Y_{t}{}^{train}=\left\{HI_{1}^{t},HI_{2}^{t},\ldots ,HI_{k}^{t}\right\}\\ Y_{t}{}^{test}=\left\{HI_{k}^{t},HI_{k+1}^{t},\ldots ,HI_{m}^{t}\right\} \end{array}$$
where $Y_{s}$ denotes source HI data, $Y_{t}$ denotes target HI data, $Y_{t}{}^{train}$ and $Y_{t}{}^{test}$ are target training HI data and target testing HI data, respectively, $HI_{i}^{s}$ denotes the HI value of the source at time *i*, $HI_{i}^{t}$ denotes the HI value of the target HI at time *i*, *n* is the length of the source HI data, k is the length of target training HI data, and m is the length of target HI data.

The source dataset $Y_{s}$ is first used to pre-train n individual LSTMi (i=1,2,…,5) models. Subsequently, the knowledge learned from the source dataset is employed to help complete the targeted task through transfer learning. As shown in Figure 5, the learned parameters $\theta^{i}$ of trained ${\mathrm{LSTM}}_{i}$ models are transferred to LSTM-${\mathrm{TL}}_{i}$ models of the target. Finally, the training data of target $Y_{t}{}^{train}$ are utilized to fine-tune the parameters of LSTM-${\mathrm{TL}}_{i}$ models to fit the targeted task.

In parameters transfer, the parameters learned from the source domain can be effectively utilized to optimize the target model. It uses the knowledge learned from the source domain to improve the learning task in the target domain. This improves the RUL prediction accuracy of LSTM models that have a small set of training data.

#### 3.2.2. Ensemble LSTM-ETL

After parameter transfer, a single LSTM model may have the risk of poor performance. In order to avoid this problem, ensemble learning is utilized to overcome this drawback. Therefore, the main goal of ensemble learning is to decrease the risk of creating a learning algorithm with poor performance. The framework of the LSTM-ETL model is shown in Figure 6.

In previous research, a widely used optimization method with random gradient descent (SGD) and momentum has been used to minimize the expected generalization error between the output and the real value. This maximized algorithm accuracy and robustness. The cost function of the model is defined as
(9)$$C_{R}=C+\lambda \theta \left(w\right)=\frac{1}{2}\overset{T}{\underset{t=0}{\sum }}\|y_{t}-{\bar{y}}_{t}{\|}_{2}^{2}+\frac{\lambda }{2}w^{T}w$$
where C is the cost function of the model, λ is the L2 regularization factor of the model, θ(w) is the regularization term that relates to the weight, $y_{t}$ is the output value of the model at time *t*, and ${\bar{y}}_{t}$ is the label value of the model at time *t*.

The learned parameters of n LSTM models are transferred to construct n LSTM-TL models. Subsequently, the LSTM-ETL model is established by using a BP neural network to integrate n individual LSTM-TL models. Detailed steps of the ensemble strategy are described as follows:(1)Construction of LSTM-ETL model: The output of n LSTM-TL models is integrated by a BP neural network. This layer is used to assign the model weights $w_{i}{}^{e}$ to $y_{t}{}^{i}$ and to calculate the ensemble predictive output $y_{t}{}^{ensemble}$ of n LSTM-TL models based on weights $w_{i}{}^{e}$ such that $y_{t}{}^{ensemble}=f\left(\sum_{i=1}^{n}y_{t}{}^{i}\cdot w_{i}{}^{e}+b\right)$.(2)Training algorithm of LSTM-ETL model: To obtain the health degradation model of the system, the LSTM-TL model is used to realize system tracking, with the HI values of *k* (*k* = 5) consecutive time points $x_{t}=\left[HI_{t-k+1},HI_{t-k+2},\ldots ,HI_{t}\right]$ as input and the HI value of the prospective time points $y_{t}=\left[HI_{t+1}\right]$ as labels. In the next iteration, the output and input of the model are $x_{t+1}=\left[HI_{t-k+2},HI_{t-k+3},\ldots ,HI_{t+1}\right]$ and $y_{t+1}=\left[HI_{t+2}\right]$, respectively. In each iteration, the input of the model consists of the second HI value to the last HI value of the input in the previous iteration as well as the prediction output from the previous iteration [21]. By the method described above, n LSTM-TL models contain a set of parameters that are adjusted. To further update these parameters and the parameters of the secondary model (BP), we use an optimization method with random gradient descent (SGD) and momentum to update the parameters (weights, w, and biases, b) to minimize the expected generalization error between the output of LSTM-ETL and the real value. By the predicted value $y_{t}$ and the real value ${\bar{y}}_{t}$, the parameters of the feedforward neural network can be obtained by error back-propagation of the cost function.(3)Prediction of LSTM-ETL model: After the degradation LSTM-ETL model has been obtained, the future HI can be achieved by introducing the prediction of the previous step into the LSTM-ETL model. In each iteration, the inputs of n LSTM-TL models are made up of the last time *k* − 1 input from the previous iteration and the last time-step prediction output from the previous iteration. For example, if the input of the previous time is $x_{t-1}=\left[HI_{t-k+1},HI_{t-k+2},\ldots ,HI_{t}\right]$, then the input of the current time is $x_{t}=\left[HI_{t-k+2},HI_{t-k+3},\ldots ,y_{t-1}{}^{ensemble}\right]$. The prediction results of n LSTM-TL models are integrated into a final result by the BP neural network. When the final result is below the predefined threshold for the first time, the model is terminated to obtain the RUL value. Finally, the formula for calculating the final RUL result is as follows:(10)$$x_{t}=\left[HI_{t-k+2},HI_{t-k+3},\ldots ,y_{t-1}{}^{ensemble}\right]$$
(11)$$y_{\left(t\right)}{}^{i}=Relu\left(w_{yh}{}^{i}h_{t}{}^{i}+b_{y}{}^{i}\right)$$
(12)$$y_{t}{}^{ensemble}=f\left(\overset{n}{\underset{i=1}{\sum }}y_{t}{}^{i}\cdot w_{i}{}^{e}+b\right)$$
(13)$$RUL_{predict}=\{k\left|y_{\left(k\right)}^{ensemble}\geq y_{threshold}\right\}$$
where $w_{yh}$ is the weight between a hidden layer and an output layer, the vector $b_{y}$ is the bias parameter of the output layer, $y_{\left(k\right)}$ denotes the predicted value of the model in step *k*, and $y_{threshold}$ denotes the predefined failure threshold.

## 4. Experiment Verification

### 4.1. An Experiment System

Experimental data are provided by the Key Laboratory of Education Ministry for Modern Design and Rotor-Bearing System [21], which has been widely employed to research RUL prediction. In the experiments, two accelerometers are positioned on the horizontal and vertical directions of the tested bearings; one is mounted on the horizontal axis and the other one is mounted on the vertical axis. The data of tested bearings were acquired on the experimental platform shown in Figure 7. Sampling frequency was set to 25.6 kHz. A total of 32,768 datapoints (i.e., 1.28 s) were recorded for each sampling, and the sampling period was equal to 1 min. To avoid redundancy, the horizontal vibration was only employed to verify the algorithms in this paper.

In the experiments, there were three different operating conditions: (1) 2100 rpm and 12 kN, (2) 2250 rpm and 11 kN, and (3) 2400 rpm and 10 kN. The full cycle data data of Condition 2 is shown in Figure 8. The last two datasets were selected in this paper to demonstrate the effectiveness of the proposed method. Bearing2_1 data and bearing2_2 data from Condition 2 were regarded as the training dataset for constructing the HI model of the target domain. Bearing3_1 data and bearing3_2 data from Condition 3 were regarded as the training dataset for constructing the HI model of the source domain.

### 4.2. Health Indicator Construction

According to [18], the structure of DBN is composed of two RBMs, where the number of hidden layers is set to three. The size of the input layer is 3000. The size of the first hidden layer, the second hidden layer, and the final hidden layer is 1000, 500, and 100, respectively. The input layer of SOM is 100, and the output layer is 1. After the structure of DBN and SOM are determined, the HI is constructed following the process shown in Figure 3.

In order to verify the performance of the proposed construction method, the RMS [23], PCA [24], and DBN-SOM methods are compared for bearing3_1. As shown in Figure 9, compared with the constructed HI curves of the traditional methods, the degradation curve constructed by the DBN-SOM method is smoother and has better monotonicity. To further illustrate the effectiveness of the DBN-SOM method, the correlation and monotonicity are used to evaluate the constructed HI. The former represents the linear relationship between the constructed HI and the sampling time point. The latter evaluates the increasing or decreasing trend of the constructed HI over time. The formula is as follows:(14)$$Corr=\frac{\left|\sum_{t=1}^{T}\left(F_{t}-\overset{ˇ}{F}\right)\left(l_{t}-\overset{ˇ}{l}\right)\right|}{\sqrt{\sum_{t=1}^{T}{\left(F_{t}-\overset{ˇ}{F}\right)}^{2}{\left(l_{t}-\overset{ˇ}{l}\right)}^{2}}}$$
(15)$$Mon=\left|\frac{Num\;of\;dF>0}{T-1}-\frac{Num\;of\;dF<0}{T-1}\right|$$
where $F_{t}$ represents the HI value of the sample at time *t*, $l_{t}$ represents the time value of the sampling point, dF represents the differential of the HI sequence, and T represents the sampling points of the whole bearing cycle.

Table 1 shows the monotonicity and correlation results of the HI curves for the RMS, PCA, and DBN-SOM methods. It can be seen that the HI performance of the DBN-SOM method on most bearings is significantly better than the two methods used for comparison. Finally, the constructed HIs are shown in Figure 10. It can be seen that the HI constructed by the DBN-SOM method can effectively reflect the state of the bearing.

The process of bearing degradation is usually divided into two stages. In the first stage, the bearing is in a healthy state with no risk of bearing failure. In the second stage, the bearing is at risk of failure. If the training data from the first stage are included to train the model, the irrelevant time-series data will interfere with the construction of the model. Therefore, partial data without the trend of degradation is ignored. In the second stage of the constructed HI value, the first 20% of the data are used for training data, and the last 80% of the data are used for validation data.

### 4.3. RUL Prediction

The main objective of the proposed method is to improve the RUL prediction accuracy of LSTM in a small set of training samples. To achieve this objective, five LSTM models first are pre-trained through the source HI data, and then the LSTM-ETL model is retrained by the target HI data. Finally, the model parameter settings are shown in Table 2.

To demonstrate the performance of the LSTM-ETL model for RUL prediction, the proposed method is compared with the LSTM, LSTM-TL, and SVM methods. The trained set is used to update the RUL model; the value of RUL can be calculated when the predicted HI reaches the failure threshold, which is set 1.0.

To compare the performance of the LSTM-ETL model, two methods to calculate the error of prediction results are applied to evaluate the performance of the RUL prediction, which are defined as follows:(16)$$Er_{i}=\frac{ActRUL_{i}-PreRUL_{i}}{ActRUL_{i}}\times 100\%$$
(17)$$\mathrm{MAE}=\frac{1}{n}\overset{n}{\underset{i=1}{\sum }}\left|Er_{i}\right|$$
where $ActRUL_{i}$ denotes the true RUL of the *i*th bearing data, $PreRUL_{i}$ denotes the predicted RUL of the *i*th bearing data, $Er_{i}$ denotes the average error of *i*th bearing data, and MAE denotes the mean absolute error of all bearing data.

### 4.4. Results and Discussion

The results of prediction are shown in Figure 11, in which the evaluation result of the proposed method is compared with other methods for accuracy. As shown in Figure 11, in bearings2_5, the predicted RULs based on the SVM, LSTM, LSTM-TL, and LSTM-ETL models are 18,100 s, 16,600 s, 13,300 s, and 9990 s, respectively. The corresponding actual RUL is 8570 s. The error of the LSTM-ETL model is less than the error of the SVM, LSTM, and LSTM-TL models. As mentioned before, smaller error indicates more-accurate prediction. Thus, it is proved that the proposed method can improve model performance with a small set of training samples.

In order to further illustrate the performance of the proposed method, five bearings under Condition 2 are selected as the target bearings to be predicted. The LSTM-ETL, LSTM-TL, LSTM, and SVM models are employed to predict RUL. Table 3 shows the results of the RUL predicted by the LSTM, LSTM-TL, LSTM-ETL, and SVM models in terms of both the Er (percentage error) and MAE (mean absolute error). The following two significant conclusions can be drawn from the results listed in Table 3.
(1)Based on the overall MAE, the performance of the LSTM-ETL method is better than that of the LSTM, LSTM-TL, and SVM methods. From the perspective of Er in individual bearings, the proposed method has lower error for all four bearings except bearing2_4. When there is insufficient training data, the overall performance of the LSTM-ETL method is better than that of the other methods.(2)LSTM-ETL shows a MAE of 31.89%, which shows that the proposed method can precisely predict the value of RUL with a small set of data. Comparing with the MAEs of LSTM (63.39%) and SVM (66.07%), it can be concluded that both transfer learning and ensemble learning are conducive to the LSTM model’s higher prognostic accuracy. The MAE of the LSTM-ETL model is 13.54% higher than that of the LSTM-TL model, which shows that ensemble learning can effectively improve the performance of RUL prediction models for single-transfer learning.

## 5. Conclusions

In this study, an HI based on DBN and SOM was proposed and constructed to enhance RUL prediction accuracy of bearings. During the construction of the HI, the process of feature extraction and feature fusion did not need artificial labels. Further, there was no need to determine the fault threshold (FT) based on researcher experience. Subsequently, LSTM with ensemble learning and transfer learning (LSTM-ETL) was utilized to predict the RUL of bearings. The results of the transfer learning and ensemble learning method had relatively low MAEs. The results showed that transfer learning can improve the performance of prediction models by using the knowledge learned from the source domain to improve learning in the target model, and ensemble learning can improve accuracy by integrating the results from multiple models. In addition, the proposed method performs better than the other methods, which provides an improved strategy for RUL prediction. For the negative transfer problem in transfer learning and ensemble learning, evaluation and analyzation of bearing-data information loss under different working conditions will be the topic of future work. This is a key problem that a prediction model with transfer learning and ensemble learning can be addressed through deep metric learning.

## Figures and Tables

**Figure 1 sensors-22-05744-f001:**
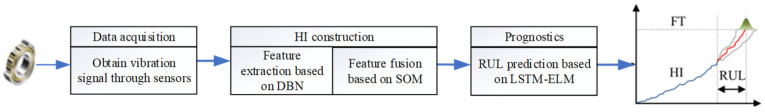
Flowchart of the proposed method.

**Figure 2 sensors-22-05744-f002:**
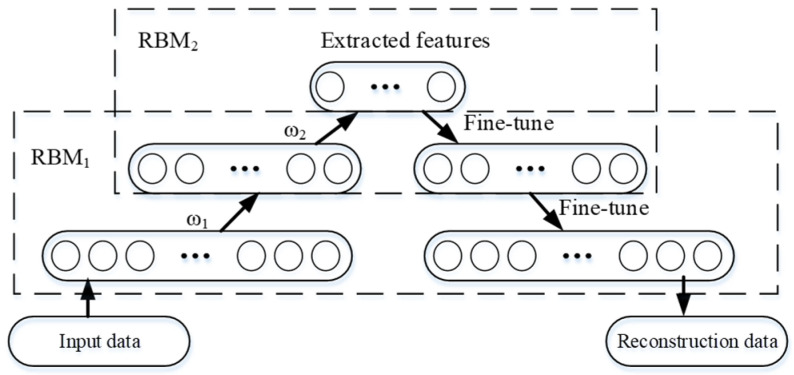
Unsupervised fine-tuning of DBN.

**Figure 3 sensors-22-05744-f003:**
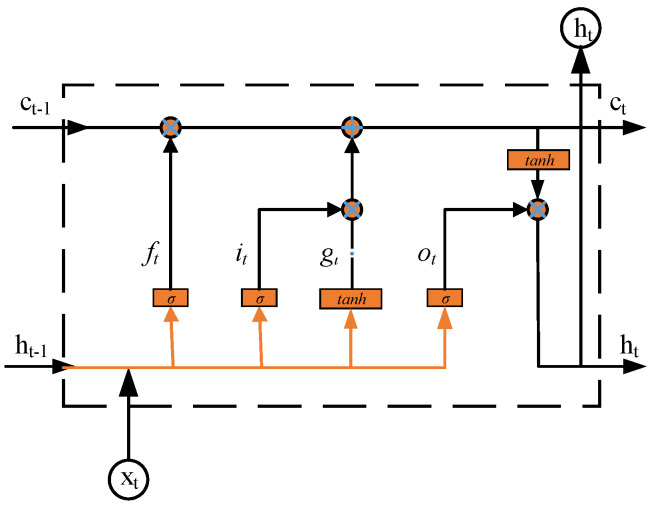
Diagram of LSTM cell.

**Figure 4 sensors-22-05744-f004:**
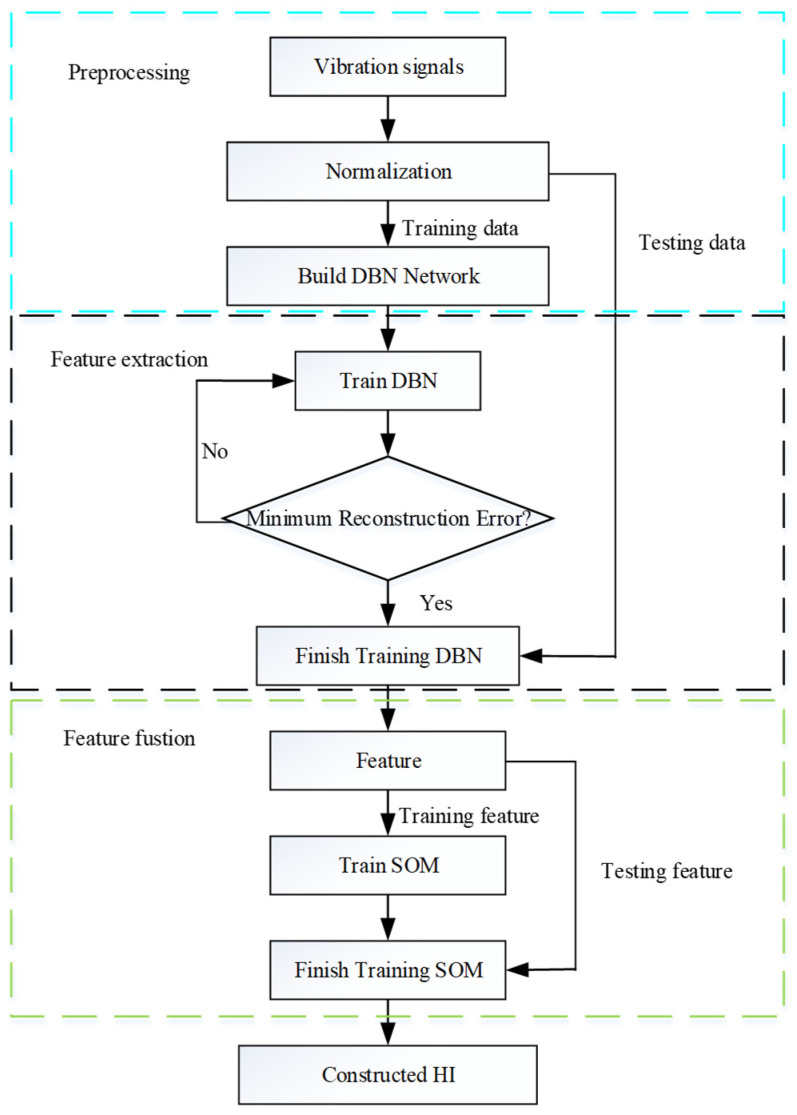
Flowchart of bearing health indicator construction.

**Figure 5 sensors-22-05744-f005:**
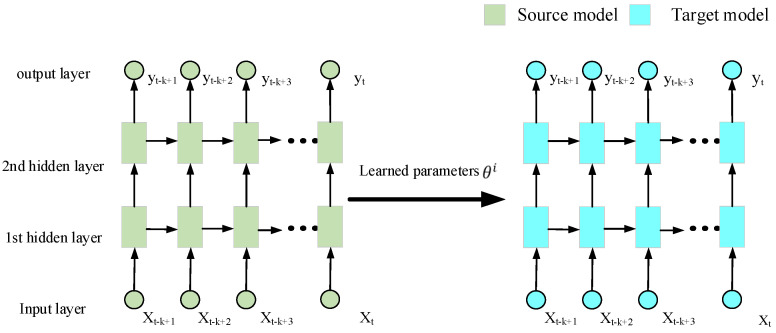
The process of transfer learning.

**Figure 6 sensors-22-05744-f006:**
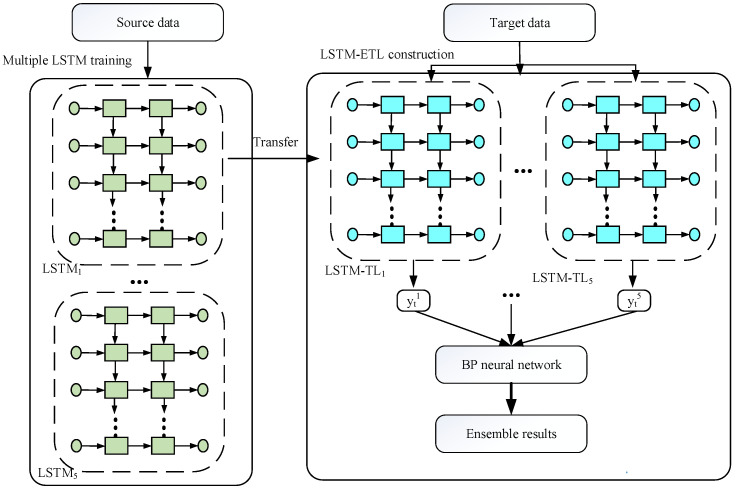
Construction of the proposed LSTM-ETL mode.

**Figure 7 sensors-22-05744-f007:**
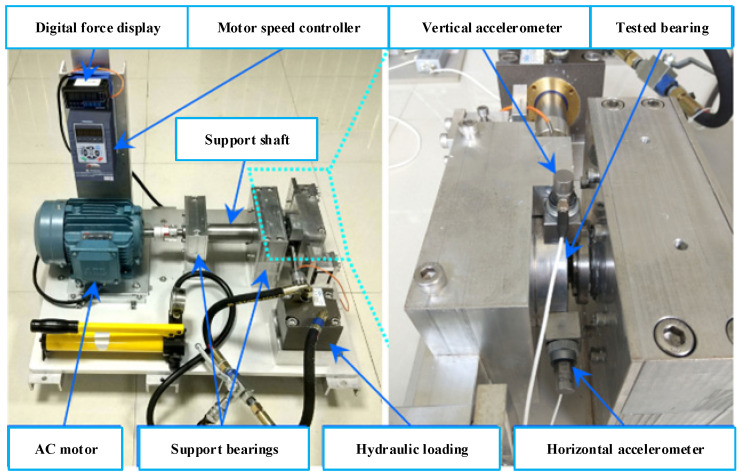
Rolling-bearing-life data-acquisition-experiment platform [22].

**Figure 8 sensors-22-05744-f008:**
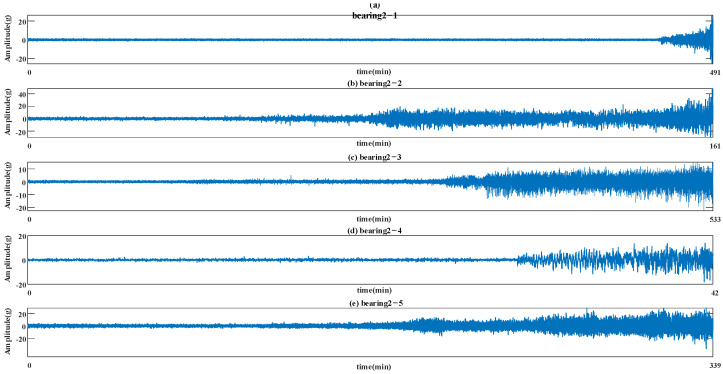
The full cycle data bearing vibration signal.

**Figure 9 sensors-22-05744-f009:**
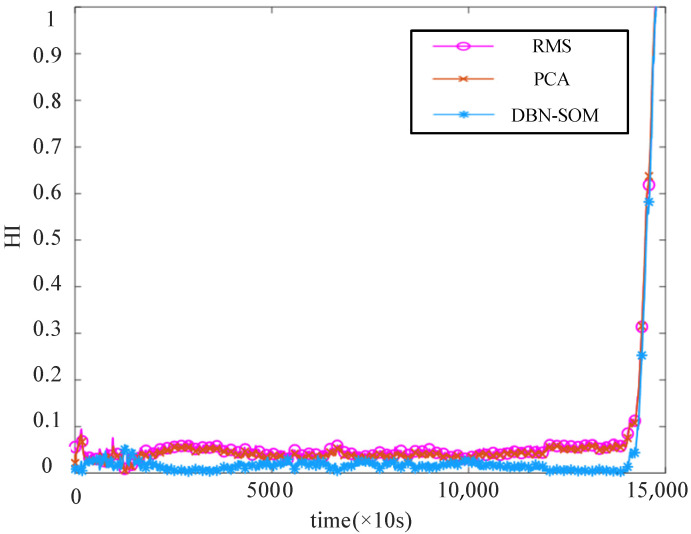
HI curve of bearing3_1 for RMS method, PCA method, and DBN-SOM method.

**Figure 10 sensors-22-05744-f010:**
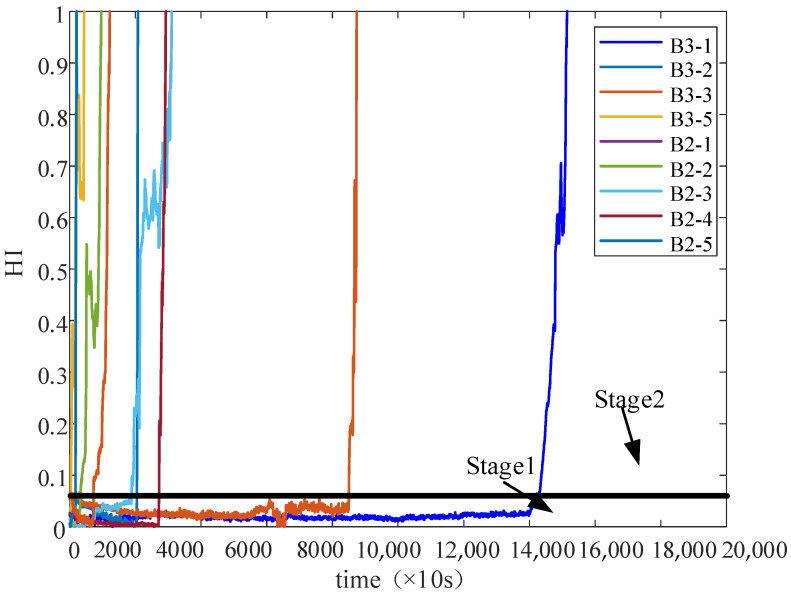
HIs of bearings.

**Figure 11 sensors-22-05744-f011:**
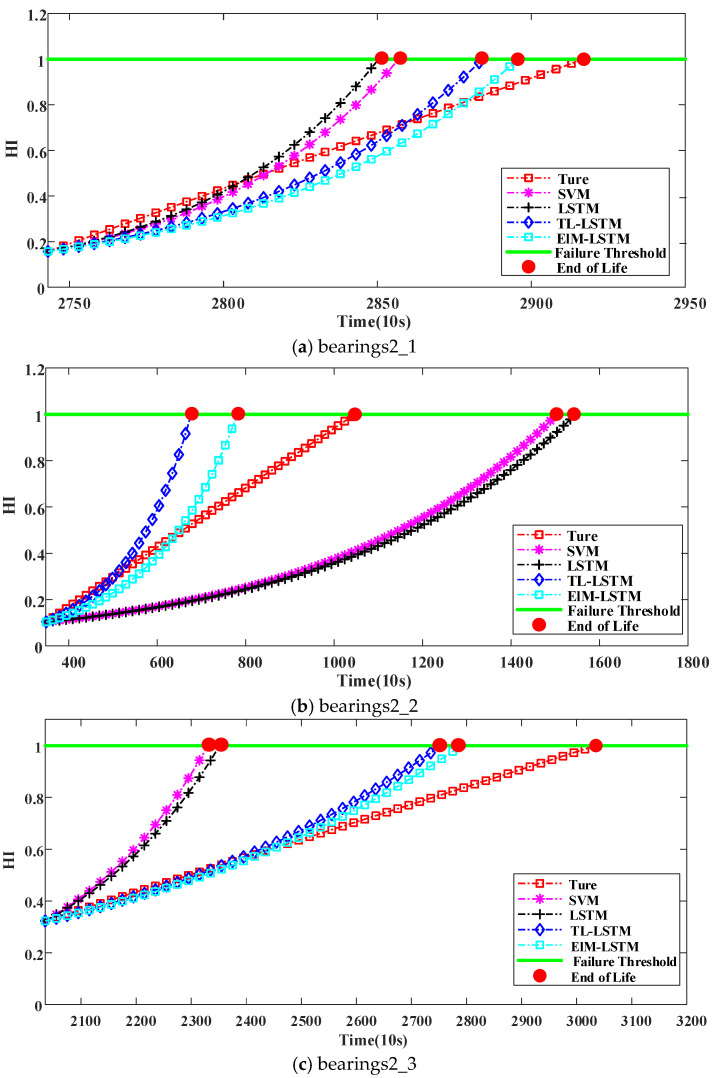
Results of RUL prediction for bearings2_1, bearings2_2, bearings2_3, bearings2_4, and bearings2_5.

**Table 1 sensors-22-05744-t001:** The monotonicity and correlation results of HI curves for RMS, PCA, and DBN-SOM methods.

Bearing	RMS		PCA		DBN-SOM	
	Cor	Mon	Cor	Mon	Cor	Mon
2_1	0.26	0.17	0.27	0.16	0.29	0.17
2_2	0.65	0.15	0.65	0.18	0.71	0.21
2_3	0.51	0.13	0.52	0.14	0.51	0.14
2_4	0.19	0.14	0.21	0.13	0.31	0.15
2_5	0.55	0.17	0.56	0.18	0.61	0.17
3_1	0.31	0.14	0.32	0.14	0.40	0.16
3_2	0.17	0.13	0.17	0.15	0.30	0.14
3_3	0.43	0.16	0.41	0.17	0.45	0.17
s3_5	0.27	0.11	0.28	0.13	0.30	0.15

**Table 2 sensors-22-05744-t002:** Parameter values used in LSTM pre-training and LSTM-ETL retraining.

Parameter	Pre-Training	Retraining
Initial learning rate	0.01	0.001
Momentum	0.9	0.9
Number of neurons	[100, 100]	[100, 100]
Number of epochs	1000	1000

**Table 3 sensors-22-05744-t003:** Prediction results.

Testing Bearing	Current Time (s)	Actually RUL (s)	LSTM-ETL Predict RUL(s)	LSTM-ETL Error (%)	LSTM-TL Error (%)	LSTM Error (%)	SVM Error (%)
2_1	27,420	1750	1530	12.57	19.36	38.18	34.32
2_2	3480	6990	4330	38.06	53.27	−70.96	−65.38
2_3	20,940	10,010	7560	24.47	28.16	68.36	70.28
2_4	1850	590	190	67.80	71.19	45.76	49.15
2_5	10,020	8570	9990	−16.57	−55.19	−93.70	−111.20
MAE				31.89	45.43	63.39	66.07

## Data Availability

The data used to support this study are available at https://biaowang.tech/xjtu-sy-bearing-datasets/ (accessed on 27 January 2021).

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
