# Peer review of "Long Short-Term Memory Neural Network with Transfer Learning and Ensemble Learning for Remaining Useful Life Prediction"

_sensors, 2022, doi:10.3390/s22155744_

Round 1

Reviewer 1 Report

1.  There are many grammatical errors in the paper, the author should improve his English expression.

2.     The aim of RUL prediction is to forecast the remaining useful life based on the data currently available to guide the equipment maintenance in advance. In the article, the authors use the data when the bearing is faulty for RUL prediction, it is confusing on the meanings of RUL prediction. 

3.     Many machine learning algorithms for the feature extraction, but their generalization are poor. It is hoped that the authors can authors to demonstrate the superiority of the proposed in the article comparing with existing algorithms.

4.     The division of data and the definition of symbol are chaotic. The authors should explain what is the raw data, HI, source data and target HI, also their construction and dimensionality. The x in 3.1 seems to refers to the raw data, but in 3.2, it is the HI.

Author Response

Reply comments are uploaded in the file.

Reviewer 2 Report

A LSTM model combined transfer learning and ensemble learning is proposed for the RUL prediction of rolling bearings. A method that uses DBN and SOM to construct HI indicator can be obtained from raw sensor data. But there are still some problems in this paper, which is listed as follows.

 1. There are lack of necessary explanations of the role and effects of adopted SOM.

2. There is no accuracy comparison for the HI indicators constructed by unsupervised construction method. For example, it is necessary to make a comparison between artificially constructed HI indication and the proposed method.

3. Is the constructed HI indicators used as the label value for RUL prediction, and if so, what is the purpose of RUL prediction?

4. Not detailed introduction to the training data and test data.

5. The input data in ensemble learning is consistent, how to ensure the reduction of model variance?

6. The English expression of the paper needs to be improved.

7. The layout of pictures and formulas needs to be improved.

To sum up, the construction of the HI index of the paper lacks the confirmation of accuracy, the RUL method has certain problems, and the experimental setting is not clear, so it is not recommended to accept it.

Author Response

(The authors gave the same response as above.)

Reviewer 3 Report

Recommendation: Major Revision

 Comments:

1.  Why Er (error results of percentage error) and MAE (mean absolute error) were chosen as predictors for the evaluation.

2.       How to select training data and validation data.

3.       Please provide some direction for future research.

4.     Please cite more recent references from The Sensors to show the relevance of your study for the journal.

Author Response

Reply comments are uploaded in the file

Reviewer 4 Report

This paper presents the approach to predict remaining useful life. The scope of this paper is actually, especially due to the currently long waiting times for the delivery of parts (subassemblies).

·       Do the authors apply batch normalization or dropout techniques in the proposed LSTM?

·       What kind of the interpolation technique (if any?) was used for datasets?

·       Please define more precisely and detailed the additional contribution of the research to the recent state of the research field. The discussion must include the results obtained in comparative analysis.

Author Response

(The authors gave the same response as above.)

Round 2

Reviewer 1 Report

I have no comments and suggestions

Reviewer 3 Report

Recommendation : Accept